# The neural correlates of shared and individual experience
Peter Coppola [1,2] ✉, Adrian M. Owen[3], David K. Menon [1,4], Lorina Naci [5,6,7] &
Emmanuel A. Stamatakis [1,2,7] ✉

We set out to explore the neural correlates of individual-specific experiences. We propose an approach through which we compute individual-specific dynamics of functional connectivity states. These dynamics do not require estimation of common states across individuals and can be directly related to dynamic behavioural ratings of subjective experience. To this end, we leverage a unique functional magnetic resonance imaging dataset where subjects listened to an engaging naturalistic story while awake and under different levels of anaesthesia, altering or abolishing conscious experience. We find that this method can detect correspondences between neural and subjective dynamics. We then show that the dynamics of the default mode network are more dissimilar between participants during awareness compared to unconsciousness and therefore may tend to underlie more personal experiences of the story. On the other hand, the auditory and posterior dorsal attention networks show higher inter-subject similarity in consciousness compared to unconsciousness and suggest that the dynamics of these networks support more "generalisable" experiences of the story. We further characterise individual-specific brain dynamics by showing that they are associated with higher complexity in consciousness, whilst conversely, brain dynamics underlying shared experience become less complex during the conscious experience of the story.

A common intuition is that human experience is deeply personal, so much so that we cannot even know if it is communicated faithfully. In the words of William James, 'Absolute insulation, irreducible pluralism, is the law'[1].

Most cognitive neuroscience research up to date has focused on estimating common neural states[2–5]. Computational techniques used to this end include independent/principle component analysis, k-means clustering, hidden Markov modelling[4,6–8], inter-subject correlation of activity[9–12], multivariate pattern analysis and the related representational similarity analysis[2,3,13–15]. Such techniques have, in fact, been useful to characterise the dynamics of different pathological and consciousness states in terms of functional networks (i.e. coordinated brain activity)[4,7,15–17]. Further, they facilitate the comparison of functional network dynamics of different individuals in different states of consciousness through an estimated common reference point. However, consciousness is intrinsic to the specific individual subject[18–21]. Averaged, 'common-denominator', extrinsically-defined, network states may only permit the investigation of limited aspects of neural function[22–25]. Studies of individual differences in subjective perception[12,23,26–30] are rarely conducted in naturalistic tasks in a way that preserves individual-specific dynamics[15,16,31,32]. Yet, it is highly plausible that individual-specific neural variability (e.g. via environment or genetics) constitutes a fundamental mechanism, through which adaptive and maladaptive perception and behaviour arises[1,25,33–36].

Thus, we set out to explore the neural correlates of experiences, which are specific to an individual. We adopt a perspective that relates individual-specific states (both neural and behaviourally-reported experiences) to themselves over time in order to delineate a dynamic landscape[17]. By comparing different states as they naturally unfold in time, we explore the common intuition that subjective experience arises from its situation within its own past and future[1,17,18,37–39].

We leverage a unique dataset[11,12,40], in which healthy individuals were scanned with fMRI while listening to a highly engaging 5-minute story and were also scanned while in resting state. These two conditions were repeated in three levels of awareness (awake, moderate and deep sedation), which permits us to tease out effects that were specific to the awareness of the story.

[1]Division of Anaesthesia, School of Clinical Medicine, Addenbrooke's Hospital, University of Cambridge, Cambridge, UK. [2]Department of Clinical Neurosciences, School of Clinical Medicine, Addenbrooke's Hospital, University of Cambridge, Cambridge, UK. [3]The Brain and Mind Institute, Western Interdisciplinary Research Building, University of Western Ontario, London, ON, Canada. [4]Wolfson Brain Imaging Centre, University of Cambridge, Cambridge, UK. [5]School of Psychology, Trinity College Dublin, Trinity College Institute of Neuroscience, Dublin, Ireland. [6]Trinity College Dublin, Global Brain Health Institute, Dublin, Ireland. [7]These authors contributed equally: Lorina Naci, Emmanuel A. Stamatakis. ✉e-mail: pc605@cam.ac.uk; eas46@cam.ac.uk

Furthermore, we used a complementary dataset where the subjective experience of narrative suspense was rated on a moment-to-moment basis[40].

We explore whether large-scale network dynamics correlate with behaviourally reported subjective experience dynamics and which networks are relevant to the more personal aspects of experience via three complementary questions.

Firstly, (1) we ask whether network dynamics map onto subjective dynamics. Specifically, we investigate whether there is an isomorphic correspondence between individual-specific similarities of neural and behaviourally reported subjective states. Put simply, the difference in subjective rating between two experiences of an individual should be proportional to the differences between the underlying neural states. Our approach redefines each temporally specific brain connectivity pattern by its similarity to its own past and future, thus emphasising the personal-intrinsic nature of the stream of consciousness[1]. Our methodology therefore results in an individual-specific dynamic landscape, which is directly comparable between individuals and the dynamics of subjective report.

We then investigate (2) the neural correlates of shared and individual experiences. Having explored which networks are relevant to the experience of the story, and considering previous work showing that functional connectivity patterns are relevant to cognitive-experiential states (e.g. refs. [41–45]), we ask whether some aspects of neural dynamics are more individual-specific during awareness of the story. We further explore our results by integrating conscious and unconscious resting state conditions in our analyses to further elucidate the relevance of brain dynamics to the content-specific awareness of a complex and engaging auditory story.

Finally, (3) following on from previous theoretical and empirical frameworks[17,46,47], we investigate if individual-specific brain network dynamics correlate with higher complexity in consciousness.

We hypothesise that the default mode network (DMN), the dorsal attention network (DAN), and the auditory networks (AUD) are relevant to the conscious experience of the story, and support individual and shared experiences. The dynamics of the DMN have been proposed to be important for consciousness due to its integrative capacity[7,47–54]. Moreover, the DMN is related to highly complex processes, such as autobiographical memory and self-related processing, context-dependent cognition, semantic processing and social cognition[32,42,47,55–62]. We therefore hypothesise that this network will be more relevant to individual-specific experiences. The DAN is

thought to mediate attention to sensory stimuli[63–66], and demonstrates a dynamic relationship with the DMN[7,41,67,68]. Crucially, it also responds to state manipulations of consciousness[7,11]. We also investigate the AUN, given its direct relevance to the modality of narrative presentation. We hypothesise the DAN and the AUD networks to underlie the more shared aspects of experience. Due to the breadth and complexity of network functions, we use different subdivisions of the DMN and DAN (Fig. 1) to enable greater precision in our analyses. For a complete picture of shared and individual-specific experiences, analyses were carried out across all canonical brain networks (total of 18) and reported in the Supplementary Information.

## Results
### Network dynamics map onto subjective dynamics
We defined individual-specific network and subjective dynamics in terms of a time-by-time similarity matrix (TTM; See 'Methods', and Fig. 2). Each temporally specific state was compared to every other state in an individual-specific manner. For the network dynamics, we first obtained a network specific (Fig. 2A) connectivity pattern for every TR (2 s; Fig. 2B), that is for every fMRI image acquired, via the difference in the estimated phase of the signal (i.e. instantaneous phase synchrony (IPS)[4,69]). We then obtained the individual-specific dynamics by investigating the similarity of each connectivity pattern to all its past and future TR-specific states using Pearson's correlation (Fig. 2C, see ref. [17]). In the resulting time-by-time similarity matrix (TTM), each column represents how a temporally-specific connectivity pattern, resulting from a temporally specific instant in the story, relates to all others (Fig. 1D). To assess the robustness of the results, we repeated all analyses across different parcellation granularities (bigger or smaller regions) and different methods (e.g. estimation of lag of bold signal, removal of auto correlation in TTMs (S1)). These are described in the 'Methods' section and Supplementary Materials; see also ref. [17] for groundwork on the method.

To directly compare brain network dynamics to the subjective rating dynamics we used data from 25 independent participants, who gave a timepoint-by-timepoint rating of how suspenseful they felt the story was whilst listening to it. We arranged these data in the same space as the brain network dynamics matrix according to the scanner sampling rate (i.e. the TR), (Fig. 2F). For each participant we took the inverse of the Euclidean distance between each temporally-specific subjective rating. This gave a

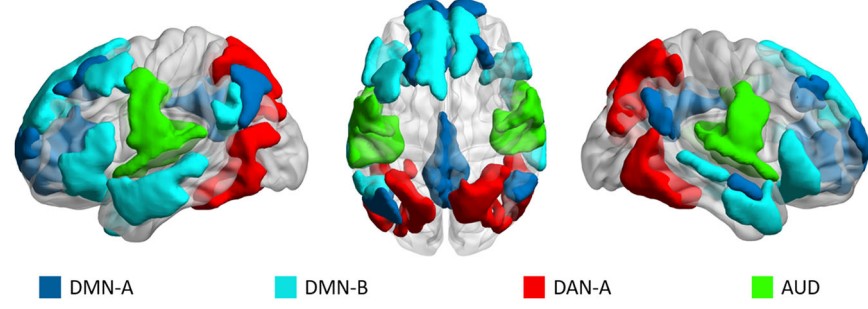

**Fig. 1 | Anatomical location of large-scale networks of interest.** We focused on four networks. These are the default mode network (DMN-A, medial, in dark blue; and DMN-B, lateral, in light blue), the dorsal attention network (DAN, only posterior part, i.e. "DAN-A", in red; see supplementary materials for DAN-B. This is not presented in the main text as it does not display strong evidence of being related to shared or individual experience), and the auditory network (AUD, in green). Note, two granularities were used to assess replicability (600 and 800 cortex parcels; see methods section. Full description of these networks and parcellations can be found elsewhere[71,72]). The DMN-A is principally composed of bilateral medial regions, including the precuneus, the posterior cingulate cortex, the posterior angular gyri, and medial prefrontal cortex. The DMN-B is composed of bilateral anterior and superior temporal regions, inferior frontal and ventrolateral regions and superior medial prefrontal cortices. DAN-A is composed of bilateral superior parietal lobe regions, temporal-occipital and temporal-parietal regions. The auditory network includes the Herschel's gyrus, planum temporale and superior temporal lobe regions bilaterally. Presented at lateral and superior views.

DMN-A    DMN-B    DAN-A    AUD

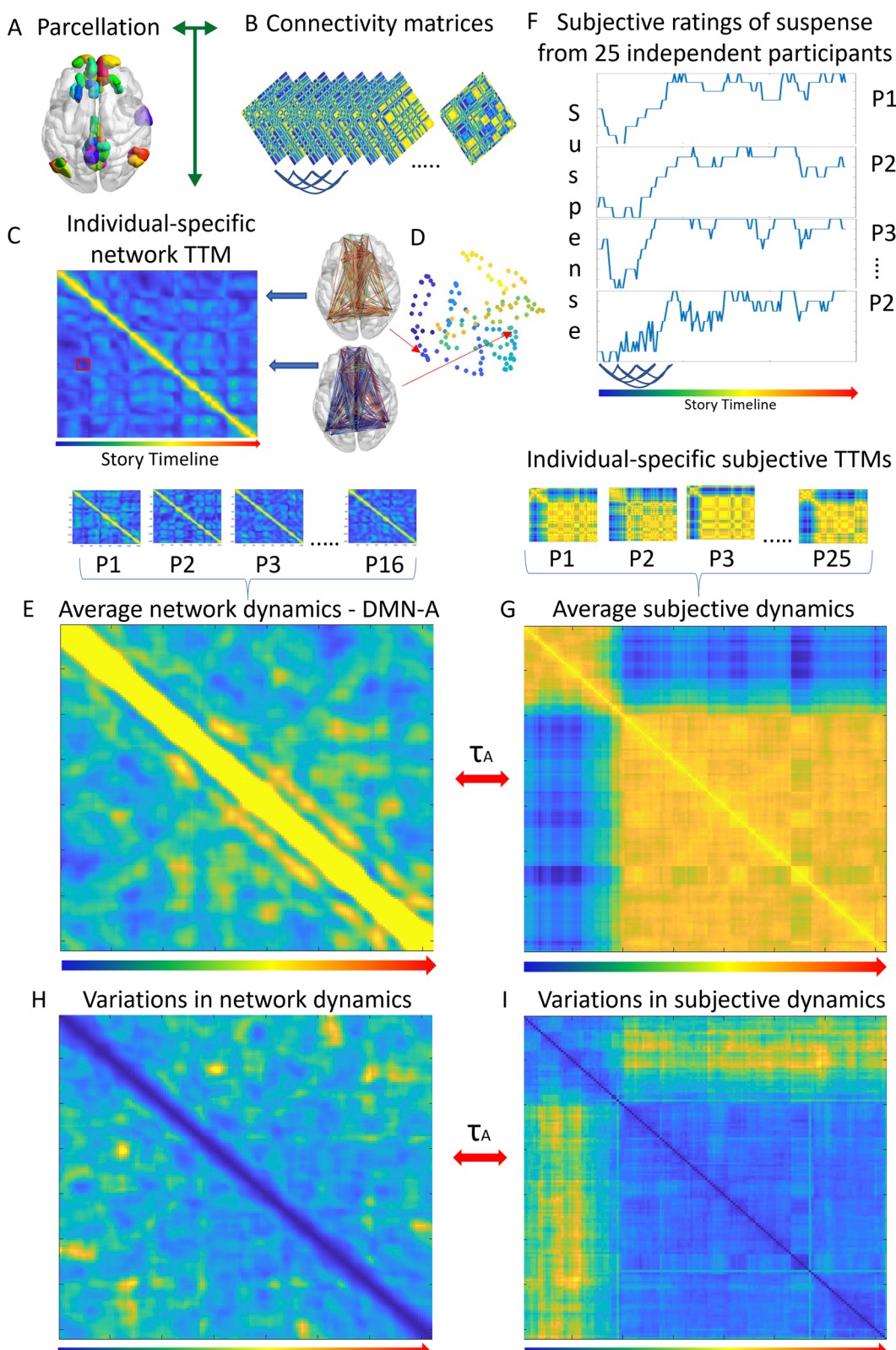

time-by-time similarity matrix which estimates subjective dynamics, i.e., a proxy of changes in experience over time or a description of 'the stream of consciousness' (Fig. 2G). Importantly, the resulting TTMs had a degree of autocorrelation which is known relevant to states of consciousness (see Coppola and colleagues[17] for how various features in TTMs change in different states of consciousness and Northoff and colleagues[48] for

theoretical elucidations). Nonetheless, to ensure that this was not unduly driving the significance of correlations between behavioural and neural TTMs, below we report *p*-values of analyses with autocorrelation removed. We plotted how average similarities between behavioural and neural TTMs changed as a function of removed autocorrelated timepoints (Supplementary Material 1). We identified points where the similarity stabilised,

**Fig. 2 | TTM construction for network and subjective dynamics.** Regional time-series were extracted via a parcellation (**A**). For comparison with subjective dynamics timeseries, the time-lag of the fMRI signal was corrected (by applying a heamodynamic response function, Wu et al.[112, 113]; alternative method in SI). Temporally specific connectivity matrices (every 2 s) (**B**) were obtained using instantaneous phase synchrony[73,75]. The upper triangle of these were vectorised and correlated (Pearson) across time-points to form an individual specific time-by-time similarity matrix (TTM; **C**). This represents individual-specific network dynamics, where each column (or row) presents a temporally specific connectivity pattern. Each cell codes the similarity (or difference) between two temporally specific connectivity patterns (**D**; note this is for representational purposes only, no analysis was conducted on the two-dimensional space shown in **D**). In **D**, each dot is a temporally specific connectivity pattern that corresponds to a temporally specific moment of the story (hotter colours indicate later in the scan/story). The position of each dot represents "where" each specific moment lies within its state space (a.k.a. phase space, spatial representation of possibilities of the system). The red square in the TTM in **C** shows the similarity values between the connectivity patterns in **D**, whilst blue arrows show the correspondence of the connectivity patterns to the columns of the TTM (**C**). The red arrows show the correspondence of the connectivity patterns to the points in the intrinsic dynamic state space (**D**). To compare the neural dynamics ($n = 16$) to behavioural ratings of subjective feelings of suspense ($n = 25$), TTMs were averaged (**E**). 25 independent participants rated their subjective feeling of suspense at the same frequency of the repetition time (2 s; **F**). By taking the inverse of the Euclidean distance between each timepoint, we obtained a TTM for subjective dynamics of suspense for each participant. Participant-specific subjective TTMs were then averaged to obtain a TTM for subjective experience (**G**). Neural and subjective dynamics (**C** and **G**, respectively) were in the same temporal space (corresponding to the "story timeline", represented by the graded arrow from blue to red) and were compared via Kendall's Tau-A ($\tau A$; Nili et al.[109]; see also 'Methods' section). Similarly, we constructed the coefficient of variance of TTMs across participants for both the neural dynamics (**H**) and the subjective dynamics (**I**). These were also compared via Kendall's Tau-A ($\tau_A$).

indicating decreased influence of autocorrelation, and report multiple comparison corrected $p$-values of the correlations with autocorrelation removed. See 'Methods' section and Supplementary Material 1 for further details.

We found that there was a correlation between the connectivity pattern (shown in Fig. 2E) and averaged behavioural ratings of subjective dynamics (shown in Fig. 2G). The DMN A ($\tau_A = 0.33$, $p < 0.0001$), and B ($\tau_A = 0.19$, $p = 0.028$), and AUN ($\tau_A = 0.21$, $p < 0.0001$) were significantly related to the subjective dynamics. The DAN-A showed a non-significant trend ($\tau_A = 0.17$, $p = 0.07$; although this was significant with an alternative method used to correct for BOLD lag; see 'Methods' and S2). All results presented were false discovery rate corrected using the Benjamini and Hochberg procedure, for all available networks using the Yeo et al., 17 canonical network definitions[70] and the subcortex[71] (S2).

### Variability in network dynamics correspond to variability in subjective rating dynamics

Above we found a correspondence between the temporal unfolding of subjective dynamics (Fig. 2G) and network dynamics (Fig. 2E). We then investigated the variations in between subjects. Specifically, we asked whether variability in neuronal dynamics maps onto the variability in subjective dynamics, in accordance with our first hypothesis.

The variability across individuals was calculated using the coefficient of variance (CV, the standard deviation normalised by the mean) on a cell-by-cell basis, for both the neural (Fig. 2H) and subjective (Fig. 2I) dynamics. The maximum CV value between individual-specific subjective dynamics was 0.43, suggesting little variation between individuals and therefore the possibility of extrapolation across samples.

We found the variation of some network dynamics consistently explained some of the variation in subjective rating dynamics. These were the DMN A ($\tau_A = 0.35$, $p < 0.001$), and the auditory ($\tau_A = 0.24$, $p < 0.001$) (false discovery rate corrected, complete results see S3). The correlations between the variability in TTMs suggest that inter-subject variations in the dynamics of these networks may track inter-subject variations in subjective dynamics.

### The neural correlates of individual and shared experience

We have so far shown that network dynamics are related to the conscious experience of the auditory story. Subsequently, we hypothesised that higher inter-subject *similarity* in a network during consciousness (compared to unconsciousness) would be evidence that the network in question tends to support shared experience. Likewise, we hypothesised that networks displaying higher *dissimilarity* in consciousness (compared to unconsciousness), would be more involved in individual-specific processes that arises with consciousness.

To this end, we first investigated whether the inter-subject similarity of the network dynamics showed an interaction between the different networks (e.g., auditory vs DMN) and the levels of awareness. The interaction effect was significant ($F = 77.27$, $p < 0.001$), indicating intersubject similarity changed according to both the network and the level of consciousness investigated. Next, we conducted planned non-parametric comparisons between the awake and the deep sedation conditions, asking whether there was higher or lower inter-subject similarity in awareness compared to deep anaesthesia.

We found that the dynamics of the DMN-A and B are more individual-specific (i.e., dissimilar between individuals) in consciousness ($Z = -6.84$, $p < 0.0001$; $Z = -8.49$, $p < 0.0001$, respectively). However, the dynamics of the AUN were more similar in consciousness ($Z = 6.62$, $p < 0.0001$, Fig. 3A, results are Bonferroni corrected), indicating that this complex stimulus engaged this network in a way that increased similarity between individuals. The DAN-A however, showed only an effect when networks were defined with smaller regions (see S4 for full results). These results suggest a dissociation of networks that support shared vs individual-specific experience during narrative understanding.

### Conscious understanding of narrative affects inter-subject similarity of network dynamics

In order to further confirm that the present approach is sensitive to the *content-specific* understanding of the narrative, we investigated whether alterations in inter-subject similarity with and without sedation could be linked directly to the presence of the story stimulus. Thus, we investigated whether the inter-subject similarities changed according to whether participants were consciously hearing the story, were conscious but with no specific content (i.e. resting), were exposed to the story during sedation and were sedated but with no stimulus.

We thus compared the similarity of the network dynamics between each participant for the rest and story conditions in the three levels of awareness (awake, moderate anaesthesia, deep anaesthesia). We built a two-way ANOVA for each network (no sedation; moderate and deep sedation for resting state and story listening conditions) and searched for interaction effects that survived Bonferroni correction across confirmation analyses (full results can be found in S4).

We found a significant interaction in the DMN B ($F = 50.11$, $p < 0.0001$), and AUN ($F = 7.02$, $p = 0.0009$; see Fig. 3B). The DAN-A and DMN-A displayed a multiple comparison corrected significance only when networks were defined with smaller regions (S5). These networks' dynamics seem to be affected both by the presence of the story and whether the subject was anaesthetised or not, suggesting that the novel approach presented is sensitive to conscious understanding of a narrative.

### Individual-specific network dynamics are more complex in consciousness

Complexity has been theorised to be fundamental for consciousness[46]. We asked whether our dissociation between dynamics that support the

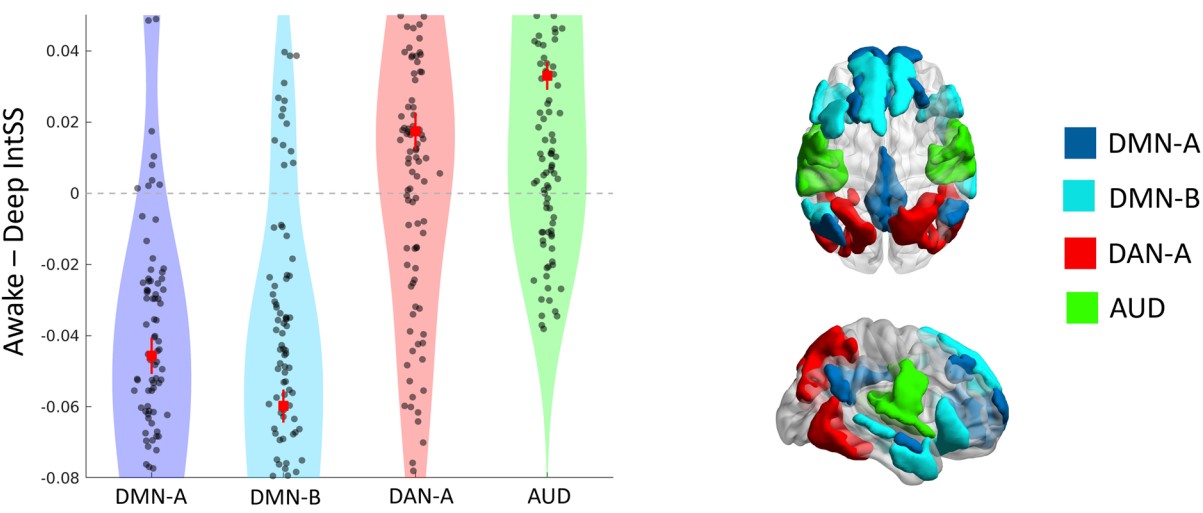

**A   Average IntSS of TTMs: Awake - Deep**

**B   Interactions of inter-subject similarity**

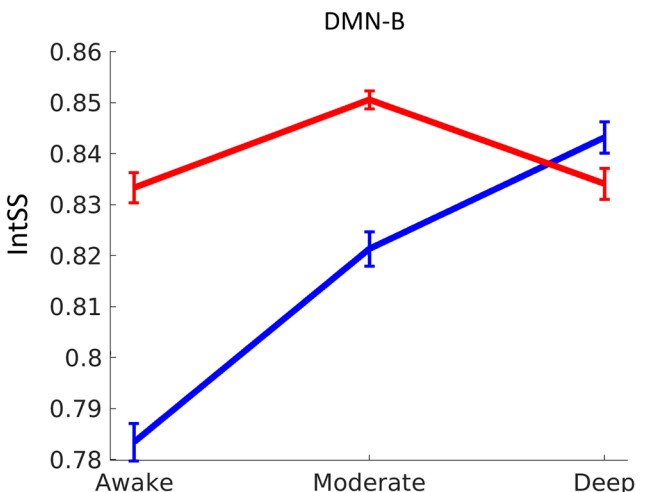

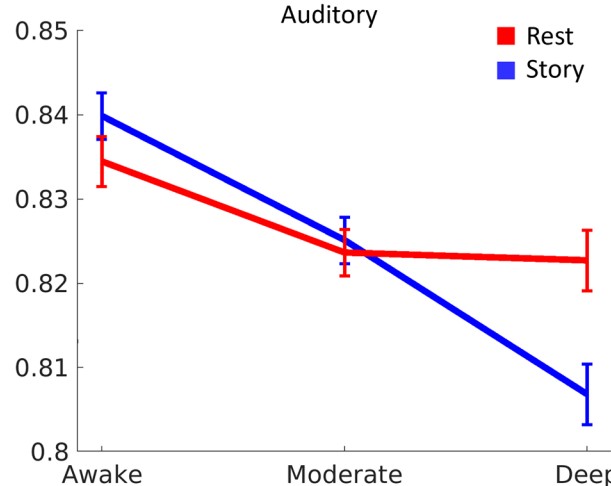

**Fig. 3 | Inter-subject similarity and dissimilarity during awareness of the narrative. A** Awake minus Deep inter-subject similarity (IntSS) of TTMs between all subjects (number of comparisons = 120). Mean represented as a red square, error bars signify bootstrapped ($n = 1000$) standard error of mean difference. Higher granularity parcellation results shown here. Right side panel shows the anatomical locations of the brain networks. Interactions (**B**) of inter-subject similarities from $3 \times 2$ ANOVA (awake, moderate, deep [presented respectively from left to right], for story [Blue] and rest [Red] conditions). These are presented for the DMN-B and auditory network. DMN default mode network, DAN dorsal attention network, AUD auditory network, IntSS intersubject similarity, Δ delta/difference.

individual and shared experience correlate with complexity. We therefore tested whether network dynamics that are individual specific (Fig. 4) tended to display more 'information' or variation in consciousness (measured via Shannon entropy). A network supporting individual-specific awareness may be assumed to display more complexity in consciousness. Thus, we expect higher complexity in individual-specific networks during consciousness. Conversely, networks that support shared experiences may be more structured in consciousness due to adaptive entrainment to the dynamic stimuli[36].

We calculated the Shannon entropy on the TTMs for each individual. We then computed whether the difference in Shannon entropy (S6) between conscious and unconscious conditions correlated with the differences in inter-subject similarity. We used all available networks ($n = 18$, shown in Fig. 4, all results available in Supplementary Materials).

The difference between awake and deep average entropy displayed a strong negative correlation (rho = −0.89; Fig. 4) with the difference between awake and deep in inter-subject similarity. Thus, network dynamics that

tended to support individual specific experiences (i.e. showed lower inter subject similarity in consciousness; Fig. 4, y-axis) displayed complex dynamics during consciousness (Fig. 4, x-axis).

## Discussion

We cannot help but think about the fundamental question of whether our experiences are shared or unique. In this study, we provide a rigorous and empirical explanation to this question by exploring the neural correlates of shared and individual-specific experiences. After finding a correspondence between neural and behavioural reported subjective dynamics, we established that the DMN is associated with individual-specific experiences. On the other hand, our findings suggest that auditory regions and the posterior DAN are related to generalisable aspects of experience.

We propose that the DAN-A is linked to processes underlying commonalities between the experiences of different individuals. As expected[11], the AUN showed evidence of being associated with the shared experience of the story (e.g. lower-level sensation; Fig. 3A, B). Conversely, given the

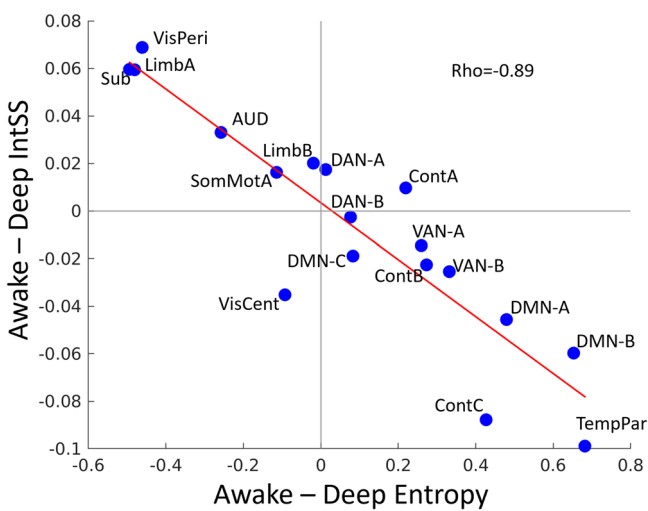

## Relationship Δ-Entropy and Δ-IntSS

**Fig. 4 | Individual specific network dynamics are more complex in consciousness.** Correlation between delta (awake-deep) intersubject similarity (intSS) and delta (awake-deep) Shannon entropy across all individual networks (Yeo et al., 17 canonical network definitions[71] and the subcortex[77]). Values for each network were averaged across individuals. TempPar temporal parietal, Cont control network, DAN dorsal attention network, Van ventral attention network, Limb Limbic network, Sub subcortical network, AUD auditory network, SomMot somatomotor network, Vis Cent central visual network, Peripheral Visual Network. See Y axis for average intSS difference between awake and deep conditions of other networks analysed. Full results presented in the Supplementary Materials.

evidence showing individual specific dynamics arising with conscious experience, we propose that the DMN-A and DMN-B are related to individual-specific experiences of the story.

Common aspects of experience across people are essential to human nature. What drives these experiential similarities between different individuals? A basis for shared experiences arises out of common phylogenic development (e.g. need to attend to the environment) and the statistical regularities of the environment (e.g. loud noises may indicate the presence of a large animal). But importantly, the commonality of experiences across humans also underlies communication and the social nature of humanity, which requires a context-dependent inter-personal generalisability between individual-specific perceptions and thoughts. On the other hand, consciousness is most easily defined in terms of subjectivity[20,21], which implies an individual-specific perspective. Although there may be a fundamental 'incommunicability' across subjective experiences[72], we have attempted to elude the problem by comparing intrinsically-defined network dynamics between individuals in different states of consciousness. In fact, the existence of individual differences in neural representational spaces are likely[16,22,23,26,31,32,73–75]. This suggests that each individual has their own neural mechanisms to represent experiences (e.g. thoughts and perceptions). We avoid issues with individual differences in representation and perception by comparing the individual-specific relationships between temporally-specific states.

Indeed, another important contribution of this work is the 'circumnavigation' of the so-called hard problem of consciousness (i.e. how does the quality of experience arise out of neural matter) (Chalmers, see p. 3 & 18[21]). This was enabled by the positioning of subjective behavioural ratings and neural data within the same internally defined dynamic space (i.e. TTM) and showing a correspondence between moment-to-moment neural interactions and subjective feelings (i.e. the rating of narrative suspense). However, the use of independent subjects for the behavioural and fMRI data in the present study weakens such inferences about neural-subjective correspondences.

The involvement of the DMN in individual-specific experience is in harmony with previous findings. This network is one of the most phylogenetically recent networks in the brain (e.g. ref. 76) and one that undergoes substantial development ontogenically (e.g. ref. 77), leaving scope for individual-specific variation. In fact, this is a network that is associated with the most complex types of cognition in humans (e.g. social, self-referential thought, mental time-travel; imagination; creativity; context-integration[32,42,56,78]), and is known to show aberrant properties in many heritable neurological and neuropsychiatric disorders (e.g. refs. 76,79).

These characteristics suggest that the DMN network is well suited to sustain highly idiosyncratic experiences. In particular, the DMN is thought to integrate long timescales of external information to information that is highly specific to the individual (e.g. memories and beliefs), to enable active, context-dependent and dynamic "sense-making"[15,16,32]. In fact, the DMN, has been consistently associated with autobiographical processing[33,80]. However, the DMN was originally found in reference to dynamics that were stimulus independent[81], and such dynamics can be found intact in individuals with reduced consciousness[82]; hence, part of the variability observed may be due to dynamics unrelated to subjective experience.

Beyond the medial aspects of the DMN (DMN-A; composed primarily of medial parietal and frontal regions; Fig. 1), we found increased individual specific dynamics in the DMN-B. This network is comprised of medial temporal regions, the angular gyri and the inferior frontal gyrus (Fig. 1). These regions are involved in language and multimodal perception[83,84] and their involvement in conscious processing of the auditory narrative is evidenced by the modest but significant interactions of inter-subject similarity in the rest and story condition as a function of consciousness (Fig. 3B). Whilst displaying somewhat diverging dynamics between subjects in consciousness; the DMN-B shows comparably high inter-subject similarity in conscious resting state, in unconscious resting state and unconscious story listening (Fig. 3B). Penfield reports that, when anterior temporal cortex regions, such as those in the DMN-B, are electrically stimulated in a conscious subject, patients report a reliving of past experiences or may give an alternative novel interpretation of the current situation[85]. He suggests that such regions are fundamental in interpreting the present, given past experiences and names such temporal regions as "the interpretative cortex"[86]. Thus, DMN-B functions may be involved in the idiosyncratic semantic and contextual understanding of the story.

The DAN has been classically associated with top-down voluntary control in the visual domain; however, a role in verbal and auditory domains has been documented[63–65]. Particularly, in this dataset, Naci and colleagues[12] showed that DAN's functional connectivity to auditory regions was altered during unconsciousness and was related to individual differences in verbal abilities. Our results, perhaps speak to the purported antithetical temporal relationship between the DMN and DAN (although see Yeshurun et al.[32]), which is known to be altered in unconscious and pathological states at rest[7,51,70], as well as in subtly different cognitive states[41]. Taken together, these results suggest that the posterior DAN-A may contribute to the ability to experience shared contents, possibly via its role in attention and working memory[63,64].

The AUN includes the Hershel's gyrus, planum temporale and superior temporal regions[87,88]. This network showed increased inter-subject similarity during the conscious processing of the story (Fig. 3A, B). The inclusion of this network in the neural substrate of shared experience is in consonance with previous research[10,12,40], which report that auditory region activity shows robust inter-subject correlations. Conversely, increased dissimilarity in unconsciousness compared to rest may be due to variations in blunting of perception and initial language processing. Our results suggest that this system's dynamics is indeed changed by the presence of consciousness, and that it likely supports some of the most generalisable processes during narrative understanding. However, its inter-subject variability did show a correlation with inter-subject variation in experience, suggesting that it may nonetheless be modulated by individual-specific processes. Furthermore, although processing of auditory signals may persist in

unconsciousness[12,89,90] (see also isolated forearm technique[91]), its function within its wider systemic context may be altered[12,92].

Our results also indicate that the network dynamics that display more inter-subject dissimilarity in consciousness ('personal experience') contained relatively more 'information' (i.e. Shannon entropy) during awareness. Conversely, networks that displayed more similarity in consciousness ('shared experience') tended to be more structured during awareness. The relative structure in 'shared experience' networks and the increased entropy of 'personal experience' networks, perhaps speaks to a dynamic balance of externally ('objective') and internally driven ('subjective') processing in the brain. This tension between more environmentally and more internally driven processes may support complex adaptive behaviours over time[34,47,93] (see Yeshurun et al.[32] for a framework).

## Methodological considerations and limitations

This study was only possible through this unique dataset. The intrinsic and personal nature of experience[1,21] is very difficult to study in modern neuroscience, which typically attempts to generalise effects of a sample to the population. In fact, individual differences are typically studied normatively between groups or using variation in specific dimensions within group[30,94–96] (there are notable exceptions to this[15,22,23,26,31]). This study offers an alternative approach which leverages intrinsic (subject-specific) dynamics, although we could not further explore individual processes due to a lack of repeated within-participant measures. This is indeed a primary limitation in the present study which weakens any inferences about subjective and neural correspondences. Future studies may obtain individual-specific data which would allow for further control analyses (e.g. using within and between participant comparisons between behavioural and neural data).

In fact, there are apparently contradictory results in previous studies which suggest that the middle temporal lobes and inferior frontal gyri are involved in shared experiences[12,14,15,40,97]. We must note the different approaches adopted, as this study focused on multivariate, intrinsically-defined dynamic connectivity states in response to naturalistic understanding of an evolving narrative, rather than lower-level voxel time courses or patterns (e.g. see Di & Biswal[97]). Although time courses may have been locked into one another, the use of a more macro state space, such as estimated by TTM (Fig. 2C, D), highlights subtleties that are invisible at the level of the time course of the single voxel (e.g., see Di & Biswal[97]). Furthermore, our approach was based on the relative differences between conscious and deeply sedated states, whilst inter-subject similarity studies typically use participants in typical states of consciousness[2,3,14,15,97,98]. Finally, although our analyses permitted a relative categorisation of a network as underlying shared or individual experiences, this coarse distinction is likely to be superseded by new techniques and paradigms which will hopefully permit a concurrent analysis at multiple fine-grain dimensions (e.g. experiential, spatial and temporal).

The use of the present parcellation[70,99] was warranted given the assignment of parcels to networks and multiple granularities available. Despite work showing a good correspondence between task and resting state networks[44,56,100], the functional segregation of the brain is likely individual and task specific (e.g. ref. 24). We avoided the use of techniques such as independent component analysis to define networks in a bottom-up manner, as parcellation assignment to the networks, and their estimation and correspondence across states of consciousness in these previous techniques may have been problematic, as explained in the introduction (see also ref. 17).

Finally, the counterbalancing of the consciousness conditions was not possible for participant safety reasons. This may have caused some repetition and mnemonic effects[89], which however, will not have been 'typical' due to the differing levels of consciousness.

A concern is that feelings of suspense may be considered only a narrow description of a human experience of a story. On the other hand, feelings of suspense are a complex composite of attentional, emotional, arousal, physiological, and cognitive components. Therefore, this measure does not allow a precise interpretation of what potential element of experience the network dynamics may be tracking.

Future studies may attempt to statistically dissociate these different elements via simultaneous and multimodal experiential and physiological dynamic measurements (e.g. skin conductance and electroencephalogram, as well as dynamic subjective reports, e.g. ref. 101). Thus, with an appropriate experimental design, variation in network dynamics can be reconducted to different dissociated physiological and experiential factors. Although experimentally tractable, the validity of such a dissociation in terms of the integration and composition of information in consciousness[1,102,103] poses interesting theoretical avenues to be further explored.

Furthermore, as is a main theme within this work, different individuals may interpret or even perceive feelings of 'suspense' differently, and the comparability of complex feelings across individuals may be called into question. This issue, a known problem in self-report measures (e.g. ref. 104), is somewhat compensated by the use of the TTMs, which redescribes the data in terms of internally-defined relative changes. Analogously to a correspondence between fMRI and subjective rating data, consistent proportionalities over time between individuals might be investigated despite differences in interpretations and scale (e.g., individual differences in perception of 'most suspenseful').

However, the subjective ratings are given from different participants to the ones from which fMRI data was acquired, limiting the possible investigating of individual aspects of experience, something that may be addressed with an ad-hoc experimental design. Furthermore, we acknowledge that the sample size is small, especially for probing intersubject similarity. Stronger evidence is needed to explore the mechanistic underpinnings of shared and individual-specific experience.

Due to the exploratory nature of this work, there are a substantial number of results pertaining to other networks covering most of the brain (total $n = 18$, the cerebellum was not investigated due to lack of spatial data). The extensive results were not conducive to accessible reporting. Nonetheless, the effects of some of the networks are notable (for example, the 'temporal parietal network' and 'control network C' in individual specific experience; and the 'subcortex', 'peripheral visual' and 'Limbic A' networks in shared experience (see Y axis of Fig. 4 for a summary)). The results pertaining to all networks analysed are presented (S2-6) and discussed in the supplementary materials (S7), should the reader be interested in further details.

## Conclusion

We have proposed a method which preserves individual specific dynamics. We find a correspondence between neural and subjective dynamics at a group level, somewhat making way on the hard problem of consciousness[21]. By investigating whether these dynamics are more similar or dissimilar between individuals in consciousness compared to unconsciousness, we have been able to propose some neural correlates of the individual-specific and shared experience of a story. We propose that whilst the DMN tends to support individual specific experience, the DAN and AUNs may support more shared aspects of experience. Despite great strides in inter-subject analyses[14,15,32] our approach highlights that consciousness is fundamentally personal[1]. Mirroring modern notions of complexity[33,47], we sustain that "you cannot step into the same river twice, for different waters flow"[105], and that experiences will vary within an individual as well as between individuals. We make one of the first efforts to explore this much more complex, yet fundamental aspect of experience.

## Methods
### Participants

These data were acquired for two previous studies (Deng et al.[40] for behavioural data and Naci et al. for fMRI data[12]). However, the approach taken and the research questions asked in this paper are novel. For the MRI propofol data, the Health Sciences Research Ethics Board and Psychology Research Ethics Board of Western University gave ethical approval. All ethical regulations relevant to human research participants were followed. Nineteen healthy, right-handed, native English speakers with no history of

neurological disorders were recruited. Three volunteers were excluded from analyses due to technical malfunctions or physiological impediments to reaching the deep sedation state in the scanner.

## Stimuli

A 5-min, plot-driven story was played to the healthy volunteers in scanner. This was the audio track from the film 'Taken' (https://en.wikipedia.org/wiki/Taken_(film)). The participants were asked to listen with their eyes closed.

A resting state scan was also acquired (8 min). During this, participants were asked to relax with their eyes closed and not fall asleep.

For the behavioural subjective ratings, 25 independent participants were asked to rate the 'suspensfullness' of the story on a moment-by-moment basis (every 2 s). Volunteers were asked to rate how 'suspenseful' the story was every two seconds (matching the fMRI TR) from 'least' (1) to 'most suspenseful' (9). The stimulus was delivered through over-ear headphones in a sound-isolated room at the Global Brain Health Institute at Trinity College Dublin. The audio clips, each two seconds long, were divided into 156 different examples. Compared to the neural data, there was one additional rating at the end, whilst the first 5 were removed to match the removed first 5 initial scans to obtain scanner equilibrium. For each clip, participants had 3 s to give a response, after which the following audio clip would commence. At the end of this behavioural experiment, participants indicated via a feedback questionnaire that the ratings did not interfere with the perceived coherence of the plot and the experience of suspense. See Deng et al.[40] for more details.

## Anaesthetic procedure

One anaesthetic nurse and two anaesthesiologists supervised the scanning session. Volunteers completed a verbal recall task and an auditory target-detection. Level of wakefulness was further assessed via an infrared camera. A Baxter AS50 (Singapore) was used to administer intravenous propofol. A computer-controlled infusion pump gave stepwise increments of dosage until the three assessors agreed that the target sedation was reached (Ramsey level 3 for moderate and Ramsey level 5 for deep anaesthesia conditions). Target propofol concentrations were predicted and maintained constant via a pharmacokinetic simulation software (TIVA trainer, European Society for Intravenous Anaesthesia, eurovisa.eu).

The preliminary propofol concentration target was 0.6 μg/ml, and step-wise increments of 0.3 μg/ml were administered. After each increment, level of wakefulness was assessed. When the target state was reached (Ramsey level 3 and 5 for moderate and deep sedation, respectively), data collection would begin. $SpO_2$ was kept above 96% to ensure oxygen titration. To ensure the safety of all subjects, scanning time was kept to a minimum. During moderate anaesthesia, the estimated average mean effect site propofol concentration was 1.99 (1.59–2.39) μg/ml and the mean estimated plasma propofol concentration was 2.02 (1.56–2.48) μg/ml. For the deep anaesthesia condition, the effect site concentration was estimated at 2.48 (1.82–3.14) μg/ml, whilst the plasma propofol concentration was 2.68 (1.92–3.44).

## fMRI acquisition

A 3 Tesla Siemens Tim Trio scanner, and a 32-channel head coil was used to acquire the neural images. Resting state (256 images) as well as the story data (156 images) were acquired. TR was 2 s, whilst TE was 30 ms. Flip-angle was 75°. There were 33 slices, with a 25% interslice gap. Resolution was 3 mm, isotropic. Anatomical T1 images were also acquired 1 mm isotropic voxels. Using an MPRAGE sequence with 9 degrees FA, a matrix of 240 × 256, TE of 4.25 ms. Acquisition time for this was 5 min.

## Preprocessing of MRI images and dynamic connectivity matrix construction

Both the story and the rest-state data were pre-processed in the same way using an in-house MATLAB script implementing SPM 12 functions (https://www.fil.ion.ucl.ac.uk/spm/).

For the functional images, the initial 5 scans were removed. We performed slice timing correction and aligned the 3D images to the mean functional image which produced the relative displacement of the functional volumes. We then normalised these to an EPI template. We created the participant-specific cerebral spinal fluid and white matter masks using the segmentation function of SPM12.

We extracted the timeseries using the SPM-based toolbox CONN. We used movement parameters to regress out movement-related data. We also removed the variance associated with the five biggest principal components obtained from participant-specific white matter and cerebrospinal fluid masks. Linear detrending was applied. A 0.03–0.07 Hz bandpass filter was also used to ensure IPS prerequisites were satisfied[69,106]. Two cortical parcellations were used to assess convergence of results. These were the 600 and 800 parcel Schaefer parcellations (Schaefer et al.[99]). These parcellations also have a direct correspondence to the 17 Yeo networks[70]. The higher granularity network definition (17 instead of 7 networks) was chosen to discriminate differential effects within canonical networks. IPS was applied to the resulting timeseries (code available from Pedersen et al.[69]; however, modified in accordance to ref. 107, as this permitted to measure anti-phase information). This involves obtaining the phase information of the BOLD timeseries via a Herbert transform. Subsequently, between region connectivity is estimated as the difference in the phases.

## Time-by-time similarity matrix (TTM) definition

The time-by-time similarity matrix (TTM) redefines each original time-point in terms of its similarity to its past and future[17]. It may be interpreted as representing an intrinsic temporal landscape (Fig. 2D). It does not require to choose any parameter and does not down sample data to clustered states. It also offers a solution to the representational problem (e.g. refs. 2,108), in that there is no direct comparison of neural states between individual (which despite being the same, may potentially underlie different experiences for different individuals, and conversely being different, may underlie similar experiences). Instead, the TTM focuses on the similarities of states within an individual. By redefining data in a temporal landscape, it permits the direct comparison of different individuals in different states of consciousness (in terms of their self-similarity over time) as well as different types of data (i.e. neural and subjective). We thus compare the loci of states visited by individuals (Fig. 2D) rather than the states themselves. This perspective is somewhat inspired from representational similarity analysis[13,109]. The distribution of TTM similarities (rather than its temporally-specific structure) has previously been used as a target for computational models[110]. This approach is also analogous to that proposed in topological data analysis[111].

The neural TTM were created from time-varying connectivity matrices for each of the 17 yeo networks plus a high granularity subcortical network[70,71,99]. Canonical network subdivisions (17 networks instead of 7) were used to enable more specific investigations (e.g., DMN-A and B; the latter more language-semantic-oriented; AUD is united with the motor network in coarser network definitions). These were constructed using IPS[69] so as to obtain a connectivity matrix for every timepoint thus obtaining a node x node x timepoint tensor (Fig. 2B). Then each of these time-specific network states was vectorised and correlated to all other connectivity matrices in the tensor (across the third dimension, i.e. time) using Pearson's correlation. This produced the time-by-time similarity matrix (e.g. Fig. 2C) where the column T represented the similarity of the node x node network at that timepoint to all other networks (each cell position representing the similarity to the corresponding connectivity timepoint of row and column). These were constructed for the Rest and Story condition. A matching number of timepoints were selected from the first half of the Rest condition (i.e. 150). Upon visual inspection, we noticed that the first and last connectivity matrices tended to be very similar in the story data. This might have been a scanner effect, an effect relating to signal processing or an effect arising from the analytical component derived from the Herbert transform underlying IPS. The exploration as to whether any one of, or a combination of these factors may explain this effect was beyond the scope of this study. We decided, therefore, to remove the first three and last three timepoints, to

ensure this high similarity between initial and final timepoints did not unduly influence results. Similarly for the subjective TTMs (described below), we averaged across participants the TTMs for each network. We also calculated the CV of the network specific TTMs across participants.

Twenty five independent participants (not subject to neural data collection) each gave a timeseries of suspensiveness ratings. To construct the behavioural time-by-time similarity matrix (Fig. 2G) for the subjective ratings of each participant (Fig. 2F), we took the Euclidean distance between all ratings, thus obtaining the distances (i.e. dissimilarities). We then subtracted the resulting values from 10, therefore obtaining similarity values from the inverse of the dissimilarity values. We organised the values into a symmetrical matrix, such that each column (or row, given TTMs are symmetrical) represented a timepoint and the cells represented similarities between timepoints (each column/row organised according to time). We thus obtained a similarity matrix for each participant, in the same temporal space as the neural TTMs, thus permitting a direct comparison between the two domains. The mean TTM was created across participants as was the CV of the time-by-time similarities. The mean TTM represents the similarity between each timepoint in terms of the average suspensiveness ratings.

The CV TTM indicates the variations in the relationships between suspensiveness rating across time, normalised by the mean. Higher values indicate higher variations in the distance between the suspensiveness ratings of those timepoints (timepoints indicated by row and column indices). The use of CV over standard deviation is warranted given the vastly different nature of the data compared (Euclidean distance of a Likert scale varying from 1 to 9 and a Pearson correlation between connectivity patterns).

## Correspondence between network and subjective dynamics

To detect a correspondence between subjective and network dynamics, we needed to account for the haemodynamic lag in the BOLD signal. We adopted two different approaches to account for the delay in the BOLD signal. The first is based on the typical approach used in event-related and block design fMRI experiments: the deconvolution of BOLD signals via a haemodynamic response function (HRF). However, given the use of a naturalistic experiment in the present study, each TR would have to be modelled as an event, therefore not permitting a clear estimation of the haemodynamic response. We employed a 'blind' deconvolution algorithm, designed to deconvolve timeseries without explicit events[112,113]. This detects abnormal spikes in the data, which are then considered events. Using these detected spikes, this toolbox estimates an HRF for each BOLD timeseries and deconvolves this. We used the canonical HRF as an initial starting point for the estimation of a region and subject-specific HRF. Results arising from this method are presented in the main text. However, we also adopted another, somewhat more coarse method, to ensure robustness of results (presented in S2 and S3). Similarly, to previous fMRI studies using naturalistic stimuli[114–116], we introduced a delay in the BOLD data so that it corresponded to the peak of the canonical HRF (which we found to be 6 s, 3TRs). Thus, for this 'lagged' analyses, we removed the first three columns of the neural TTMs, so that the subjective TTM timepoints would correspond to the neural TTM in accordance to the highest peak of the HRF. Results for the introduced lag are shown in S2 and S3. These showed, for the most part, convergent results. Inconsistent results are presented as a tilde (~) in Table S8.

Spearman's correlation was not appropriate to test the correspondence between behavioural and neural TTMs, given the substantial presence of ties in the behavioural TTMs (identical values; Fig. 2G, I). We thus compared these to the neural TTMs via Kendall's Tau-A, the script for which was obtained from the representational similarity analysis toolbox[109]. The authors of this toolbox suggest using this measure of similarity when comparing representational matrices to models with ties[109] (which is analogous to the behavioural TTM in the present manuscript). According to their analyses, the Tau-A metric (as opposed to Tau B and Tau C) is more likely to favour true models over simplified models, whilst other rank-correlation coefficients may even overestimate the performance of models[109]. We calculated significance probability via a permutation of the

columns of the subjective TTM. However, time-by-time similarity matrices typically have a degree of autocorrelation, whereby timepoints that are closer to each other naturally tend to have higher similarity (see diagonals in Fig. 2C, E, G, H, I). The variance of autocorrelation over time may contain relevant information of the neural and behavioural dynamics. However, given that statistical significance was assessed via permutation of the columns (i.e. timepoints) the autocorrelational structure that is intrinsic to time-by-time similarity matrices would not be preserved in permutations and therefore might have inflated significance. To ensure the autocorrelational structure did not drive results, we decided to run analyses with autocorrelational points removed. To estimate the amount of proximal timepoint similarity values to remove (i.e. very proximal = more autocorrelation), we plotted the similarity between the behavioural and neural TTM (averaged across networks and participants) as a function of quantity of autocorrelation removed (S1). This revealed two points at which the similarity stabilised, indicating decreased influence of autocorrelation. For the $p$-values reported in the results section, we therefore removed 10 of the most proximal timepoints. In the Supplementary Materials, we also report statistics for analyses with 24 timepoints removed, this is also a point in which similarities between behavioural and neural TTMs stopped decreasing rapidly as a function of removed autocorrelation. Note that different networks and states may have different autocorrelational properties[17,48], however, the exploration of such network specific properties was beyond the scope of this study. For further details on the relationship between autocorrelation removal and similarity see S1. Thus, 10 and 24 autocorrelated timepoints were removed and the behavioural TTMs were then permuted 10000 times for each comparison with the neural TTMs. FDR correction was applied (within parcellation granularity). In the main text we report the correlation value (Tau A) between the neural and behavioural of intact TTMs (without removing autocorrelation values), as the values along the diagonals (similarity between proximal timepoints) may encode relevant information (e.g. a rapid transition between states or a prolonged lingering)[17]. However, given the presence of autocorrelation may have inflated $p$-values in the permutation analyses, in the main text we report the $p$-values obtained from removing 10 proximal timepoints (see S1, S2 and S3). Full results for all networks can be found in S2 and S3 (For mean and CV TTM correlations, respectively). The 10 and 24 autocorrelational removal showed a reasonable degree of convergence across different granularities. The TTMs' lower triangle was always vectorised before they were compared.

## Intersubject TTM similarity of brain networks

For the neural analyses, the test statistic was the similarities between each pair of participants (values = $16 \times 15/2 = 120$). These similarities were obtained by Fisher-z transforming the neural TTM and correlating the vectorised lower triangle using Pearson's correlation between participants. For the neural analyses, we used Bonferroni correction for each network and looked for convergence across granularity (600 and 800 parcels for the whole cortex (Schaefer et al.[99]). Inferential statistics regarding differences between two populations were all conducted using non-parametric sign-rank tests in MATLAB.

To further assess the relevance of the task vs rest effects across different levels of awareness, we built a two-way ANOVA for each condition (using the 'anovan' MATLAB function). We investigated specifically the interaction effect between level of awareness and condition (story vs rest). Each interaction p-value was corrected for multiple comparison using Bonferroni correction. Full results for this are available in S5. Surprisingly, in the DMN-B and other networks (see S4), we found higher inter-subject similarity (IntSS) in the rest condition compared to the story condition (Fig. 3B). Upon further exploration, we found that there was generally higher temporal autocorrelation in the rest condition, potentially reflecting smoother (subjective and network) dynamics. Higher autocorrelational structure will result in higher IntSS (see S1). Nonetheless, we show that the neural TTMs have task-specific information via the neural to behavioural TTM correspondence and the interaction analysis (Fig. 3B).

Furthermore, the relatively short timeframe of the story would have increased the proportion of autocorrelated timepoints relative to temporally distal timepoints. The timepoints of the rest condition were reduced to match those of the story. In fact, the DMN activity was found to be particularly related to rest conditions[81], and such intrinsic dynamics of the resting state[68], which can be complex to interpret[32,117,118] were used to define these networks[70,99] and may partially explain such a high IntSS. Although an interesting result in its own right, however, further investigation of task and content-dependent autocorrelation would require further data was beyond the scope of this study which focused on the neural correlates of shared and individual experience.

Finally, we tested whether there was an interaction effect between the inter-subject similarities within each propofol condition and the different networks. The interaction was of interest given that we predicted that in some networks inter-subject similarity would increase with awareness, whereas in others it would decrease. We thus built a $3 \times 18$ ANOVA (propofol conditions x all networks [$n = 18$]) for each level of granularity. We placed a linear predictor for the conditions, to assess whether inter-subject similarity would scale with the level of awareness. We used a continuous predictor (coded as 1:1:3) for the propofol condition and a categorical variable for the networks ($n = 18$). The use of a continuous linear predictor for the scaling of inter-subject similarities between conditions may be considered problematic (however see refs. 51,119). However, this was deemed sufficient for the present analysis, given that planned non-parametric post-hoc comparisons were conducted and such a predictor could ask the question whether there was an effect that scaled with levels of awareness. In line with the hypothesis that some networks would show increasing inter-subject similarity with levels of consciousness, and other networks would show decreasing similarities with levels of consciousness, the interaction effect was the primary outcome variable in this test.

The post hoc comparisons were conducted by comparing the control awake inter-subject similarities to the deep sedation inter-subject similarities in each of the networks. These were again conducted for both parcellation granularities, and using the non-parametric Wilcoxon signed rank test. Probability values for population inference were corrected for multiple comparisons using the Bonferroni correction. In line with the present hypothesis that some networks would show higher inter-subject similarities in consciousness, whilst other networks would show lower inter-subject similarities in consciousness (presumably corresponding to generalisable and individual-specific experience respectively), particular attention was placed on the sign of the test statistic.

Finally, we wanted to investigate whether the TTMs had a more complex distribution of similarity values in consciousness or unconsciousness. We were particularly interested in whether the complexity of TTMs relates to whether there is more or less intersubject similarity in comparing consciousness to unconsciousness. We thus applied the entropy function implemented in MATLAB to the TTMs for each participant during awake and deep anaesthesia conditions. We then contrasted the resulting values using the Wilcoxon signed rank test. Full results presented in S6.

## Statistics and reproducibility

As noted above, for the neural ($n = 16$) to subjective ($n = 25$) correspondence analysis, we removed autocorrelation (see above or S1). We then averaged the matrices across individuals and vectorised them (only upper or lower triangle as they are symmetrical). We compared the matrices using Kendall's Tau-A. We then created a null distribution by permuting the columns and rerunning the correlation with autocorrelation removed.

For the CV analysis, we calculated the for every cell of the matrix and we repeated the analysis as described above.

For the intersubject similarity, we tested the similarity between each pair of participants (values = $16 \times 15/2 = 120$). These similarities were obtained by Fisher-z transforming the neural TTM and correlating the vectorised upper or lower triangle of the matrix using Pearson's correlation between participants. To test differences, we used the non-parametric signed-rank test in MATLAB.

For the entropy to intersubject similarity correlation analysis, we calculated the Shannon entropy using MATLAB's entropy function. We subtracted the resulting value within participants (awake versus deep) and then averaged all values across participants to obtain one value per network. We averaged across all participants and subtracted the inter-subject similarity values (described above) to obtain one value per network. We correlated these values for the networks using Spearman's Rho.

Full methodological details can be found above.

### Reporting summary

Further information on research design is available in the Nature Portfolio Reporting Summary linked to this article.

## Data availability

The datasets analysed during the current study are available upon reasonable request. Data to reproduce the figures in the main text are available from https://github.com/Peter6789/Neural-correlates-of-individual-and-shared-experience.

## Code availability

Code used for this article is available from https://github.com/Peter6789/Neural-correlates-of-individual-and-shared-experience.

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

## Acknowledgements

The Canada Excellence Research Chairs program (215063) (A.M.O.); The Canadian Institute for Advanced Research (RCZB/072 RG93193) (A.M.O., D.K.M., and E.A.S.); Cambridge Biomedical Research Centre and NIHR Senior Investigator Awards (D.K.M.); The British Oxygen Professorship of the Royal College of Anaesthetists (D.K.M.); Cambridge Trust (P.C.); The L'Oreal-Unesco for Women in Science Excellence Research Fellowship (L.N.); The Stephen Erskine Fellowship, Queens' College, University of Cambridge (E.A.S).

## Author contributions
Conceptualisation: P.C., E.A.S. and L.N. Methodology: P.C. and E.A.S. Software: P.C. Validation: P.C. and E.A.S. Formal analysis: P.C. Investigation: D.K.M., A.M.O., E.A.S. and L.N. Resources: E.A.S., L.N., D.K.M. and A.M.O. Data curation: L.N. Writing—original draft: P.C., E.A.S. and L.N. Writing—reviewing and editing: P.C., E.A.S., L.N., D.K.M. and A.M.O. Visualisation: P.C. Supervision: E.A.S., L.N. and D.K.M. Project administration: E.A.S., L.N., D.K.M. and A.M.O. Funding acquisition: L.N., A.M.O., E.A.S., D.K.M. and P.C.

## Competing interests
The authors declare no competing interests.
