## [Transparent Peer Review file · Communications Biology]

The Neural Correlates of Shared and Individual Experience

Corresponding Author: Dr Peter Coppola

Version 0:

Reviewer comments:

Reviewer #1

(Remarks to the Author)

This study explores an important and timely topic in cognitive neuroscience: how brain network dynamics relate to shared versus individual-specific experiences during narrative processing, and how these dynamics are altered by anesthesia. The authors introduce a novel analytic approach involving time-by-time similarity matrices (TTMs) and attempt to link network dynamics with subjective ratings of suspense, while leveraging a rich dataset with naturalistic stimuli and different levels of consciousness. While this study addresses an exciting and difficult question with novel methodology, I believe several interpretive and methodological weaknesses currently limit the strength of its conclusions.

Major Concerns:

1. The manuscript lacks detailed validation of the analytic choices made. For example, the handling of autocorrelation in TTM matrices and the use of Kendall's Tau-A need more robust justification.
2. The authors draw conclusions from correlations across independent datasets—subjective ratings from one group and neural data from another. This indirect approach weakens the inference about neural-subjective correspondences.
3. The surprisingly high inter-subject similarity (IntSS) in the rest condition compared to the story condition is counterintuitive and suggests potential methodological issues or artifacts. This finding requires clearer explanation or control analyses.
4. The inference that the default mode network (DMN) supports more "personal" experiences because of greater inter-subject dissimilarity is not fully convincing. The DMN is known to exhibit robust spontaneous activity, even under deep anesthesia, and this does not necessarily equate to rich conscious content. The authors should acknowledge that spontaneous DMN activity may not map directly onto changes in conscious experience. The observed variability might reflect intrinsic dynamics unrelated to moment-to-moment phenomenology.
5. A core claim of the paper is about individual-specific experience. However, no individual-level subjective data were collected from the fMRI participants. Using an entirely separate sample for suspense ratings undermines the claim of linking individual neural dynamics to individual experience. Suspense ratings, though temporally resolved, capture only a narrow slice of the multidimensional nature of conscious experience. The manuscript would benefit from a more nuanced discussion of the limitations of using "suspense" as a proxy for experience dynamics.
6. The correlations between neural and subjective TTMs, while statistically significant in some networks, are not strong. Given the complexity of the constructs, stronger evidence is needed to support causal or mechanistic claims.

Reviewer #2

(Remarks to the Author)

This paper leverages a highly unique data set (fMRI data collected during story listening while undergoing general anesthesia) to investigate brain signatures of shared (operationalized as inter-subject similarity) and individual experience. The data set is rare and the results are interesting. I have a few methodological concerns.

First, not all analyses mention correction for multiple comparisons across brain regions and networks. For instance, in Section 3.1.1, there is no mentioning of multiple comparison correction. This might have been an omission during the writing, but if no correction was carried out this would be a concern given the large number of regions/networks tested.

The results shown in Fig. 3C should probably be removed from the paper, given that they are not significant after multiple-

comparison correction and the scatter plots are very unconvincing. Overall, there is a concern about statistical power since $N = 25$ is a fairly small sample size to probe inter-subject correlation.

Finally, a design concern is that subjective rating was only given for “suspense”, which is a pretty narrow aspect of the subjective experience when listening to stories.

Reviewer #3

(Remarks to the Author)

This is a very interesting manuscript, leveraging a truly unique dataset. The authors attempt to tease apart the different networks underlying individual-specific experiences, vs. shared experiences. For this, they examine a dataset which includes resting state fMRI and listening to a story, in 3 levels of consciousness, including two states of moderate and deep anesthesia. They compare the shared network dynamics of the DMN-A, DMN-B, DAN and the auditory network across participants, between rest and story, and across the 3 levels of consciousness. As DMN shows reduced intersubject similarity (IntSS0) during consciousness compared to anesthesia, they conclude that it supports more diverse, individual experiences. This is in contrast to DAN and the auditory networks, where consciousness prompts greater IntSS which the authors argue is indicative of the greater shared experience processing that characterizes these networks.

Overall, I really enjoyed the paper. The analyses are cleverly designed, the limitations of the dataset are acknowledged, and the findings are interesting. A few of the results are a bit overstated though. In particular, the scatterplots in Fig 3C, on which the authors rely to claim a correlation between the level of sedation in the moderate anesthesia condition, and IntSS, are not very convincing. Most participants have very similar reaction times, and appear to be similarly affected by the sedation, and their IntSS ratings span pretty much the full range. Then, there are three outliers, with much slower reaction times. Those three drive the correlations reported in the paper, not so much because their IntSS scores are so different, but mainly because their RTs are so different. In fact, if you remove them, it seems like you would get either no correlation or the opposite correlation from that reported in the paper, for both DAN and DMN.

The difference between the IntSS scores across the three levels of consciousness, while apparently robust and certainly interesting, is also overstated. The range of IntSS for the DMN-B shown in Fig 3B is between 0.78 to 0.85. The interaction between states of consciousness and the presence of the narrative is no doubt significant, but it is still a small effect. Even during listening to the story while fully awake, participants show very strong correlations (0.78). This does not be characterized as “widely different dynamics between subjects in consciousness”, as the authors state in p.12 of the discussion. It would be more accurate to say that there was a modest but significant reduction in synchronization between subjects during consciousness.

The networks the authors chose to focus on in the main text appear intuitive, but from the supplementary figures it is clear that some other networks in fact show even greater modulation in IntSS between consciousness and sedation. Why was the choice made to put those in supplementary, and not even mention them in passing in the main text? This creates a bit of a distortion in the understanding of what is really involved in subject-specific processing, or potentially could change the interpretation of the meaning of this modulation completely.

Incidentally, while I certainly agree that individual differences in neural representations exist, MVPA decoding within vs. across individuals is not a good example. The differences in decoding accuracy likely depend on the specific fine-grained anatomic / functional organizational quirks between individuals. When brains are hyperaligned, you can actually get superior cross-subject than within subject decoding, and other decoding techniques which train on large datasets also show superior decoding on new participants than decoding algorithms trained purely on subject specific data, highlighting the role of shared, rather than subject-specific information (see for instance Taschereau-Dumouchel 2018)

At the bottom of p.10: “Thus, network dynamics that tended to support individual specific experiences (i.e., showed higher inter subject similarity in consciousness; fig. 4, y-axis) displayed complex dynamics during consciousness (fig. 4, x-axis).” Shouldn't it read i.e. showed lower inter subject similarity in consciousness?

Version 1:

Reviewer comments:

Reviewer #1

(Remarks to the Author)

I appreciate the authors' thorough response to my concerns. The revision is clearly improved and well executed.

Best,
Zirui

Reviewer #3

(Remarks to the Author)

The authors have addressed all of my concerns.

Response to reviewers

We would like to thank the reviewers for their engagement and comments. We have endeavoured to answer the points raised and feel the manuscript has improved significantly. In particular, following the reviewer's comments, we acknowledge and discuss the limitations and provide more balanced interpretations of the results.

In this document:

in **bold** are the reviewer comments.

In **red** are changes to the text.

Indented are direct citations from the text.

Reviewers' comments:

Reviewer #1 (Remarks to the Author):

This study explores an important and timely topic in cognitive neuroscience: how brain network dynamics relate to shared versus individual-specific experiences during narrative processing, and how these dynamics are altered by anesthesia. The authors introduce a novel analytic approach involving time-by-time similarity matrices (TTMs) and attempt to link network dynamics with subjective ratings of suspense, while leveraging a rich dataset with naturalistic stimuli and different levels of consciousness. While this study addresses an exciting and difficult question with novel methodology, I believe several interpretive and methodological weaknesses currently limit the strength of its conclusions.

We would like to express our gratitude to the reviewer for their engagement and comments. Below is a point-by-point response to the reviewer's comments.

Major Concerns:

1. The manuscript lacks detailed validation of the analytic choices made. For example, the handling of autocorrelation in TTM matrices and the use of Kendall's Tau-A need more robust justification.

We would like to thank the reviewer for pointing out a lack of clarity.

Regarding the autocorrelation we have inserted information in two different places following the reviewer's comments. Firstly, in the results section, we have endeavoured to make the analytical choices more explicit. This is succinct as the

methods section, and the supplementary materials explicate all the methodological details. Please see the modifications of the result section below.

Lines 145-155:

Importantly, the resulting TTMs had a degree of autocorrelation which is known relevant to states of consciousness (see Coppola and colleagues¹ for how various features in TTMs change in different states of consciousness and Northoff and colleagues² for theoretical elucidations). Nonetheless, to ensure that this was not unduly driving the significance of correlations between behavioural and neural TTMs, below we report p-values of analyses with autocorrelation removed. We plotted how average similarities between behavioural and neural TTMs changed as a function of removed autocorrelated timepoints (supplementary material 1). We identified points where the similarity stabilised, indicating decreased influence of autocorrelation, and report multiple comparison corrected p-values of the correlations with autocorrelation removed. See methods section and supplementary materials 1 for further details.

Please see the following from the methods section that explains in more detail the analytical choices made in relation to the autocorrelation, with modifications to enhance clarity.

Lines 649-680:

We calculated significance probability via a permutation of the columns of the subjective TTM. However, time-by-time similarity matrices typically have a degree of autocorrelation, whereby timepoints that are closer to each other naturally tend to have higher similarity (see **diagonals in figs 2C, 2E, 2G, 2H and 2I**). The variance of autocorrelation over time may contain relevant information of the neural and behavioural dynamics. However, given that statistical significance was assessed via permutation of the columns (i.e., timepoints) the autocorrelational structure that is intrinsic to time-by-time similarity matrices would not be preserved in permutations and therefore might have inflated significance. To ensure the autocorrelational structure did not drive results, we decided to run analyses with autocorrelational points removed. To estimate the amount of proximal timepoint similarity values to remove (i.e., very proximal = more autocorrelation), we plotted the similarity between the behavioural and neural TTM (averaged across networks and participants) as a function of quantity of autocorrelation removed (S1). This revealed two points at which the similarity stabilised, **indicating decreased influence of autocorrelation**. For the p-values reported in the results section we therefore removed 10 of the most proximal timepoints. In the supplementary materials we also report statistics for analyses with 24 timepoints removed, this is also a point in which similarities between behavioural and neural TTMs stopped decreasing rapidly as a function of

removed autocorrelation. Note, that different networks **and states** may have different autocorrelational properties^{1,2}, however the exploration of such network specific properties was beyond the scope of this study. For further details on the relationship between autocorrelation removal and similarity see S1. Thus, 10 and 24 autocorrelated timepoints were removed and the behavioural TTMs were then permuted 10000 times for each comparison with the neural TTMs. FDR correction was applied (within parcellation granularity). In the main text we report the correlation value (Tau A) between the neural and behavioural of intact TTMs (without removing autocorrelation values), as the values along the diagonals (similarity between proximal timepoints) may encode relevant information (e.g., a rapid transition between states or a prolonged lingering)¹. However, **given the presence of autocorrelation may have inflated p-values in the permutation analyses**, in the main text we report the p-values obtained from removing 10 proximal timepoints (see S1, S2 and S3). Full results for all networks can be found in S2 and S3 (For mean and CV TTM correlations respectively). Further information can be found in supplementary material 1 in relation to the removal of autocorrelation.

Regarding Kendall's Tau, this is a metric that has proven to be particularly robust and appropriate for the present scenario. In the figure 2 caption, we now explicitly refer any interested reader to the methods section.

Lines 182-183:

*via Kendall's Tau-A (τ_A ; Nili et al., 2014³; **see also methods section**)*

We have now modified the methods section to provide clarity on our methodological choices.

Lines 641-649:

Spearman's correlation was not appropriate to test the correspondence between behavioural and neural TTMs, given the **substantial** presence of ties in the behavioural TTMs (identical values; fig. 2G and 2I). We **thus** compared these to the neural TTMs via Kendall's Tau-A, the script for which was obtained from the representational similarity analysis toolbox³. **The authors of this toolbox suggest using this measure of similarity when comparing representational matrices to models with ties³ (which is analogous to the behavioural TTM in the present manuscript). According to their analyses, the Tau-A metric (as opposed to Tau B and Tau C) is more likely to favour true models over simplified models whilst other rank-correlation coefficients may even overestimate the performance of models³.**

2. The authors draw conclusions from correlations across independent datasets—subjective ratings from one group and neural data from another.

This indirect approach weakens the inference about neural-subjective correspondences.

This is true. Regrettably, the subjective ratings data was not collected during the original study⁴, but posthumously⁵.

We note this in the limitation section, and have added the following, which we have modified to further highlight this weakness.

Lines 401-407:

This study offers an alternative approach which leverages intrinsic (subject-specific) dynamics, although we could not further explore individual processes due to a lack of repeated within-participant measures. **This is indeed a primary limitation in the present study which weakens any inferences about subjective and neural correspondences. Future studies may obtain individual specific data which would allow for further control analyses (e.g., using within and between participant comparisons between behavioural and neural data).**

Following the reviewer's comment, we also note the following.

Lines 455-457:

However, the subjective ratings are given from different participants to the ones from which fMRI data was acquired, limiting the possible investigating of individual aspects of experience, something that may be addressed with ad-hoc experimental design.

To ensure that this weakness is clearly acknowledged we furthermore note this in the beginning of the discussion section.

Lines: 327-332:

This was enabled by the positioning of subjective behavioural ratings and neural data within the same internally-defined dynamic space (i.e., TTM), and showing a correspondence between moment-to-moment neural interactions and subjective feelings, (i.e., the rating of narrative suspense). **However, the use of independent subjects for the behavioural and fMRI data in the present study weakens such inferences about neural-subjective correspondences.**

3. The surprisingly high inter-subject similarity (IntSS) in the rest condition compared to the story condition is counterintuitive and suggests potential methodological issues or artifacts. This finding requires clearer explanation or control analyses.

The reviewer is right; this was rather surprising.

We ran several analyses to investigate what might be the cause of this. The analyses indicated that in general the rest condition has higher autocorrelation. This

may partially explain inflated correlation values that relate to specific dynamics that characterise the rest condition, rather than content specific dynamics. We refer the reviewer to the following paragraph in the methods section:

Lines 696-703.

Surprisingly, in the DMN-B and other networks (see S4) we found higher inter-subject similarity (IntSS) in the rest condition compared to the story condition (Fig. 3B). Upon further exploration we found that there was generally higher temporal autocorrelation in the rest condition, potentially reflecting smoother (subjective and network) dynamics. Higher autocorrelational structure will result in higher IntSS (see S1). **Nonetheless, we show that the neural TTM have task-specific information via the neural to behavioural TTM correspondence and the interaction analysis (fig 3B).**

As above, we also note that different networks and states may have different autocorrelation properties which further may complicate the picture.

Lines 667-668:

Note, that different networks **and states** may have different autocorrelational properties^{1,2}.

Furthermore, the relatively low amount of data (less than 5 minutes) may have further increased the IntSS due to the higher ratio of timepoints displaying autocorrelation to distal timepoints. We have added this point

Lines 704-706:

Furthermore, the relatively short timeframe of the story would have increased the proportion of autocorrelated timepoints relative to temporally distal timepoints. The timepoints of the rest condition were reduced to match those of the story.

In fact, these networks were found during resting state. Hence the TTMs may be particularly sensitive to the dynamics that defined these and may be in part driving higher INTss. We have added this point

Lines 706-709:

In fact, the DMN activity was found to be particularly related to rest conditions⁶, and such intrinsic dynamics of the resting state⁷, which can be complex to interpret⁸⁻¹⁰ were used to define these networks^{11,12} and may partially explain such a high IntSS.

Our previous study developing the present technique (Coppola et al.,2022¹) showed that this autocorrelated information is relevant to the state of consciousness and very robust to methodological contingencies. Future studies may look further into how autocorrelation relates to the content of consciousness. We note this in the methods section.

Lines 709-712:

Although an interesting result in its own right, however, further investigation of task **and content-dependent** autocorrelation **would require further data** was beyond the scope of this study **which focused on the neural correlates of shared and individual experience.**

4. The inference that the default mode network (DMN) supports more "personal" experiences because of greater inter-subject dissimilarity is not fully convincing. The DMN is known to exhibit robust spontaneous activity, even under deep anesthesia, and this does not necessarily equate to rich conscious content. The authors should acknowledge that spontaneous DMN activity may not map directly onto changes in conscious experience. The observed variability might reflect intrinsic dynamics unrelated to moment-to-moment phenomenology.

The reviewer makes an important point.

Although the interaction analyses and the correspondence between average behavioural and neural data do suggest an involvement in content specific processes, we cannot rule out that these effects are driven by spontaneous activity. We have inserted the following in the discussion.

Lines 344-347:

However, the DMN was originally found in reference to dynamics that were stimulus independent⁶, and such dynamics can be found intact in individuals with reduced consciousness¹³; hence part of the variability observed may be due to dynamics unrelated to subjective experience.

We have also moderated claims the abstract in light of this:

Lines 26-28:

We then show that the dynamics of the default mode network are more dissimilar between participants during awareness compared to unconsciousness **and therefore may tend to underlie more personal experiences of the** story.

We also make note the coarseness of the distinction between individual specific and share experience in the following:

Lines 418-422:

Finally, although our analyses permitted a relative categorisation of a network as underlying shared or individual experiences, this coarse distinction is likely to be superseded by new techniques and paradigms which will hopefully permit a concurrent analysis at multiple fine-grain dimensions (e.g., experiential, spatial and temporal).

5. A core claim of the paper is about individual-specific experience. However, no individual-level subjective data were collected from the fMRI participants. Using an entirely separate sample for suspense ratings undermines the claim of linking individual neural dynamics to individual experience. Suspense ratings, though temporally resolved, capture only a narrow slice of the multidimensional nature of conscious experience. The manuscript would benefit from a more nuanced discussion of the limitations of using "suspense" as a proxy for experience dynamics.

As above, we agree with the reviewer and acknowledge that suspense ratings being collected from an independent sample is an issue. The interaction analysis was performed to increase confidence that at least part of these dynamics was related to being exposed to the stimulus during consciousness. To further increase confidence in our results, we investigated correlations between subjective ratings and the network dynamics that show more inter subject dissimilarity in consciousness compared to unconsciousness. The low coefficient of variance reported also might indicate the possibility of extrapolating between samples.

Lines 203-205:

The maximum CV value between individual-specific subjective dynamics was 0.43, suggesting little variation between individuals and therefore the possibility of extrapolation across samples.

Nonetheless, this remains a limitation of the study as noted in various parts of the manuscript (copied in response above).

We do agree that suspense ratings are limited. The encouragement of the reviewer in engaging in more nuanced considerations of the limitations has led to several interesting points and potential future directions for consciousness science. We have added these to the limitations section and copy the modifications below in their entirety.

Lines 434-454:

A concern is that feelings of suspense may be considered only a narrow description of a human experience of a story. On the other hand, feelings of suspense are a complex composite of attentional, emotional, arousal, physiological, and cognitive components. Therefore, this measure does not allow a precise interpretation of what potential element of experience the network dynamics may be tracking.

Future studies may attempt to statistically dissociate these different elements via simultaneous and multimodal experiential and physiological dynamic measurements (e.g., skin conductance and electroencephalogram, as well as dynamic subjective reports, e.g.,¹⁴). Thus, with an appropriate experimental design, variation in network dynamics can be reconducted to different

dissociated physiological and experiential factors. Although experimentally tractable, the validity of such a dissociation in terms of the integration and composition of information in consciousness¹⁵⁻¹⁷ poses interesting theoretical avenues to be further explored.

Furthermore, as is a main theme within this work, different individuals may interpret or even perceive feelings of “suspense” differently, and the comparability of complex feelings across individuals may be called into question. This issue, a known problem in self-report measures (e.g.,¹⁸), is somewhat compensated by the use of the TTMs, which redescribes the data in terms of internally-defined relative changes. Analogously to a correspondence between fMRI and subjective rating data, consistent proportionalities over time between individuals might be investigated despite differences in interpretations and scale (e.g., individual differences in perception of “most suspenseful”).

6. The correlations between neural and subjective TTMs, while statistically significant in some networks, are not strong. Given the complexity of the constructs, stronger evidence is needed to support causal or mechanistic claims.

We agree that we cannot make mechanistic claims. Beyond the modification of the abstract (copied above), we have modified language in the discussion to ensure that our results are not misinterpreted as relating to mechanistic claims. Specifically, we have changed the phrasing “involved in” to “associated with”, “related to” or “linked to”.

Lines 298-308:

After finding a correspondence between neural and behavioural reported subjective dynamics, we established that the DMN **is associated with** individual-specific experiences. On the other hand, our findings suggest that auditory regions and the posterior DAN **are related to** generalisable aspects of experience.

We propose that the DAN-A **is linked to** processes underlying commonalities between the experiences of different individuals. As expected¹⁹, the auditory network showed evidence of being **associated with** the shared experience of the story (e.g., lower-level sensation; Figs. 3A and 3B). Conversely, given the evidence showing individual specific dynamics arising with conscious experience, we propose that the DMN-A and DMN-B **are related** to individual-specific experiences of the story.

We have also added the following to the limitations section:

Lines 459-460:

Stronger evidence is needed to explore the mechanistic underpinnings of shared and individual-specific experience.

Reviewer #2 (Remarks to the Author):

This paper leverages a highly unique data set (fMRI data collected during story listening while undergoing general anesthesia) to investigate brain signatures of shared (operationalized as inter-subject similarity) and individual experience. The data set is rare and the results are interesting. I have a few methodological concerns.

We kindly thank the reviewer for their comments and engagement. Please see below responses to the reviewer.

First, not all analyses mention correction for multiple comparisons across brain regions and networks. For instance, in Section 3.1.1, there is no mentioning of multiple comparison correction. This might have been an omission during the writing, but if no correction was carried out this would be a concern given the large number of regions/networks tested.

Apologies. We acknowledge this to be an omission. We corrected for all our result. This is discussed in the methods, but we have now stated this in the results section.

Lines 207-208:

These were the DMN A ($T_A=0.35$, $p<0.001$), and the auditory ($T_A=0.24$, $p<0.001$) (**false discovery rate corrected**, complete results see S3).

Lines 227-228

However, the dynamics of the auditory network were more similar in consciousness ($Z=6.62$, $p<0.0001$, fig 3A, **results are Bonferroni corrected**).

We decided a-priori to utilise a more stringent correction method for analyses that did not use behavioural data to ensure their robustness. We chose this approach a-priori given the high dimensionality of the fMRI data and the fact that correlations to behavioural data can increase confidence in results (e.g.,²⁰).

The results shown in Fig. 3C should probably be removed from the paper, given that they are not significant after multiple-comparison correction and the scatter plots are very unconvincing.

We understand the reviewer's concern. We did hypothesise that this would show an effect. However, the results are not as strong as we might have expected.

Following the reviewer's comment, we have decided to remove this section of the results. We have also removed all the sections in the discussion and supplementary materials that relate to this result.

Overall, there is a concern about statistical power since $N = 25$ is a fairly small sample size to probe inter-subject correlation.

This is indeed another limitation of our study. Whilst the behavioural suspense ratings have $N=25$, the fMRI data has $N=16$. This was in part due to the technical, financial and safeguarding issues associated with anaesthetic administration unto Ramsey Level 5 for experimental purposes.

We have now noted this in the limitations section explicitly.

Lines 457-458:

Furthermore, we acknowledge that the sample size is small, especially for probing intersubject similarity.

We have also made the following modification to increase clarity.

Lines 174-175:

To compare the neural dynamics ($n=16$) to behavioural ratings of subjective feelings of suspense ($n=25$), TTMs were averaged (E).

Finally, a design concern is that subjective rating was only given for “suspense”, which is a pretty narrow aspect of the subjective experience when listening to stories.

It is interesting that we focused on the fact that suspense may be considered a rather complex composite. However, the reviewer is right in that suspense falls short in comprehensively describing a human experience of a narrative.

We have noted this.

Lines 434-435:

A concern is that feelings of suspense may be considered only a narrow description of a human experience of a story.

We also note that future studies might provide a more comprehensive multimodal description of how physiological dynamics map onto various experiential components. To this end we highlight the recent work by Lewis-Healey et al. that use a recently proposed method for multidimensional experiential descriptions which however relies on retrospective reporting. An alternative to which would be the use of repetitive measurements of the same stimulus which however would likely induce a significant order effect.

Lines 435-446:

On the other hand, feelings of suspense are a complex composite of attentional, emotional, arousal, physiological, and cognitive components. Therefore, this

measure does not allow a precise interpretation of what potential element of experience the network dynamics may be tracking.

Future studies may attempt to statistically dissociate these different elements via simultaneous and multimodal experiential and physiological dynamic measurements (e.g., skin conductance and electroencephalogram, as well as dynamic subjective reports, e.g.,¹⁴). Thus, with an appropriate experimental design, variation in network dynamics can be reconducted to different dissociated physiological and experiential factors. Although experimentally tractable, the validity of such a dissociation in terms of the integration and composition of information in consciousness¹⁵⁻¹⁷ poses interesting theoretical avenues to be further explored.

This is an exciting and difficult field of research. We hope the present paper may contribute to make progress.

Reviewer #3 (Remarks to the Author):

This is a very interesting manuscript, leveraging a truly unique dataset. The authors attempt to tease apart the different networks underlying individual-specific experiences, vs. shared experiences. For this, they examine a dataset which includes resting state fMRI and listening to a story, in 3 levels of consciousness, including two states of moderate and deep anesthesia. They compare the shared network dynamics of the DMN-A, DMN-B, DAN and the auditory network across participants, between rest and story, and across the 3 levels of consciousness. As DMN shows reduced intersubject similarity (IntSS0) during consciousness compared to anesthesia, they conclude that it supports more diverse, individual experiences. This is in contrast to DAN and the auditory networks, where consciousness prompts greater IntSS which the authors argue is indicative of the greater shared experience processing that characterizes these networks.

We would like to express our sincere appreciation for the reviewer's time and engagement. We have endeavoured to answer all the points raised exhaustively.

Overall, I really enjoyed the paper. The analyses are cleverly designed, the limitations of the dataset are acknowledged, and the findings are interesting. A few of the results are a bit overstated though. In particular, the scatterplots in Fig 3C, on which the authors rely to claim a correlation between the level of sedation in the moderate anesthesia condition, and IntSS, are not very convincing. Most participants have very similar reaction times, and appear to be similarly affected by the sedation, and their IntSS ratings span pretty much the full range. Then, there are three outliers, with much slower reaction times. Those three drive the correlations reported in the paper, not so much because their intSS scores are so different, but mainly because their RTs are so

different. In fact, if you remove them, it seems like you would get either no correlation or the opposite correlation from that reported in the paper, for both DAN and DMN.

Following the reviewer's comment (as well as reviewer #2 comment above), we have removed this result entirely. We as well as all mentions of this in the discussion and the supplementary materials.

The difference between the IntSS scores across the three levels of consciousness, while apparently robust and certainly interesting, is also overstated. The range of IntSS for the DMN-B shown in Fig 3B is between 0.78 to 0.85. The interaction between states of consciousness and the presence of the narrative is no doubt significant, but it is still a small effect. Even during listening to the story while fully awake, participants show very strong correlations (0.78). This does not can be characterized as “widely different dynamics between subjects in consciousness”, as the authors state in p.12 of the discussion. It would be more accurate to say that there was a modest but significant reduction in synchronization between subjects during consciousness.

The reviewer is right. We have now toned down the interpretation. We have removed the use of “widely” and “striking” in the reported paragraph.

Lines 351-356:

These regions are involved in language and multimodal perception^{21,22} and their involvement in conscious processing of the auditory narrative is evidenced by the **modest but significant** interactions of inter-subject similarity in the rest and story condition as a function of consciousness (fig 3B). Whilst **displaying somewhat diverging** dynamics between subjects in consciousness; the DMN-B shows comparably high inter-subject similarity in conscious resting state, in unconscious resting state and unconscious story listening (fig 3B).

The networks the authors chose to focus on in the main text appear intuitive, but from the supplementary figures it is clear that some other networks in fact show even greater modulation in intSS between consciousness and sedation. Why was the choice made to put those in supplementary, and not even mention them in passing in the main text? This creates a bit of a distortion in the understanding of what is really involved in subject-specific processing, or potentially could change the interpretation of the meaning of this modulation completely.

We originally attempted to report the results of all the networks involved. However, this led to an overly complex paper that was not accessible, partly due the many results and complex interpretations of the role of different networks. We therefore

decided to focus on the networks that are more amenable to interpretation given the previous literature and the nature of the task. For these networks, we had stronger a priori predictions.

Nonetheless, we appreciate the reviewers concern and agree with them. We have decided to highlight the other networks in the paper by placing names of all 18 networks in figure 4 (please note, for this figure we now use the higher granularity network definitions so that they match figure 3A). This allows the reader to get an intuitive a quick idea of the results in other networks. Should the reader be interested, they may then review all results in the supplementary materials.

Lines 275-278:

We then computed whether the difference in Shannon entropy between conscious and unconscious conditions correlated with the differences in inter-subject similarity (e.g., see fig 3C; also S6). We used all available networks ($n=18$, shown in figure 4, all results available in supplementary materials).

Lines 285-294:

Figure 4 Individual specific network dynamics are more complex in consciousness. Correlation between delta (awake-deep) intersubject

similarity (intSS) and delta (awake-deep) Shannon entropy across all individual networks (Yeo et al, 17 canonical network definitions⁸¹ and the subcortex⁸²). Values for each network were averaged across individuals. TempPar= Temporal Parietal; Cont = Control Network; DAN= Dorsal Attention Network; Van= Ventral Attention Network; Limb= Limbic network; Sub=Subcortical Network; AUD= Auditory Network; SomMot= SomatoMotor Network; Vis Cent = Central Visual Network; Peripheral Visual Network. See Y axis for average intSS difference between awake and deep conditions of other networks analysed. Full results presented in the supplementary materials.

And furthermore in the limitations section where we highlight the networks with the greatest effects.

Lines 461-469:

Due to the exploratory nature of this work, there are a substantial number of results pertaining to other networks covering most of the brain (total n=18, the cerebellum was not investigated due to lack of spatial data). The extensive results were not conducive to accessible reporting. Nonetheless, the effects of some of the networks are notable (for example the “temporal parietal network” and “control network C” in individual specific experience; and the “subcortex”, “peripheral visual” and “limbic A” networks in shared experience (see Y axis of figure 4 for a summary)). The results pertaining to all networks analysed are presented (S2-6) and discussed in the supplementary materials (S7), should the reader be interested in further details.

Incidentally, while I certainly agree that individual differences in neural representations exist, MVPA decoding within vs. across individuals is not a good example. The differences in decoding accuracy likely depend on the specific fine-grained anatomic / functional organizational quirks between individuals. When brains are hyperaligned, you can actually get superior cross-subject than within subject decoding, and other decoding techniques which train on large datasets also show superior decoding on new participants than decoding algorithms trained purely on subject specific data, highlighting the role of shared, rather than subject-specific information (see for instance Taschereau-Dumouchel 2018)

We would like to thank the reviewer for the point and the extremely interesting manuscript. Interestingly, hyper alignment seems to increase the reliability of individual differences in neural measures although diminishing effect sizes²³. Figure 1A in this paper shows the dissimilarity/similarity across individuals and how different individuals may vary in the extent of similarity. They do not show all the individual differences matrices as this was not the focus of their study.

We also note interesting work which suggests differences in hyperalignment can be used to stratify patients^{24,25}.

In the changes following the reviewer's comment, we remove references to MVPA and change the phrase in the following way:

Lines 320-321:

In fact, the existence of individual differences in neural representational spaces are **likely**^{10,23–30}.

At the bottom of p.10: “Thus, network dynamics that tended to support individual specific experiences (i.e., showed higher inter subject similarity in consciousness; fig. 4, y-axis) displayed complex dynamics during consciousness (fig. 4, x-axis).”

Shouldn't it read i.e. showed lower inter subject similarity in consciousness?

Thank you for noting this error. We have now amended it

Lines 281-283:

Thus, network dynamics that tended to support individual specific experiences (i.e., showed **lower** inter subject similarity in consciousness; fig. 4, y-axis) displayed complex dynamics during consciousness (fig. 4, x-axis).

References

1. Coppola, P. *et al.* The complexity of the stream of consciousness. *Commun. Biol.* **5**, 1–15 (2022).
2. Northoff, G. & Huang, Z. How do the brain's time and space mediate consciousness and its different dimensions? Temporo-spatial theory of consciousness (TTC). *Neurosci. Biobehav. Rev.* **80**, 630–645 (2017).
3. Nili, H. *et al.* A Toolbox for Representational Similarity Analysis. *PLoS Comput. Biol.* **10**, (2014).
4. Naci, L. *et al.* Functional diversity of brain networks supports consciousness and verbal intelligence. *Sci. Rep.* **8**, 1–15 (2018).
5. Deng, F., Taylor, N., Owen, A. M., Cusack, R. & Naci, L. Responsiveness variability during anaesthesia relates to inherent differences in brain structure and function of the frontoparietal networks. *Hum. Brain Mapp.* 2142–2157 (2023) doi:10.1002/hbm.26199.
6. Raichle *et al.* A default mode of brain function. *Proc. Natl. Acad. Sci.* **98**, 676 (2001).

7. Fox, M. D. *et al.* The human brain is intrinsically organized into dynamic, anticorrelated functional networks. *Proc. Natl. Acad. Sci. U. S. A.* **102**, 9673–9678 (2005).
8. Gonzalez-Castillo, J., Kam, J. W. Y., Hoy, C. W. & Bandettini, P. A. How to interpret resting-state fMRI: Ask your participants. *Journal of Neuroscience* vol. 41 1130–1141 at <https://doi.org/10.1523/JNEUROSCI.1786-20.2020> (2021).
9. Finn, E. S. Is it time to put rest to rest? *Trends Cogn. Sci.* **25**, 1021–1032 (2021).
10. Yeshurun, Y., Nguyen, M. & Hasson, U. The default mode network: where the idiosyncratic self meets the shared social world. *Nat. Rev. Neurosci.* **22**, 181–192 (2021).
11. Schaefer, A. *et al.* Local-Global Parcellation of the Human Cerebral Cortex from Intrinsic Functional Connectivity MRI. *Cereb. Cortex* **28**, 3095–3114 (2018).
12. Yeo, B. T. T. *et al.* The organization of the human cerebral cortex estimated by intrinsic functional connectivity. *J. Neurophysiol.* **106**, 1125–1165 (2011).
13. Boly, M. *et al.* Intrinsic brain activity in altered states of consciousness: How conscious is the default mode of brain function? in *Annals of the New York Academy of Sciences* vol. 1129 119–129 (Blackwell Publishing Inc., 2008).
14. Lewis-Healey, E., Tagliazucchi, E., Canales-Johnson, A. & Bekinschtein, T. A. Breathwork-induced psychedelic experiences modulate neural dynamics. *Cereb. Cortex* **34**, (2024).
15. Koch, C., Massimini, M., Boly, M. & Tononi, G. Neural correlates of consciousness: Progress and problems. *Nat. Rev. Neurosci.* **17**, 307–321 (2016).
16. James, W. *The Principles of Psychology*. 4004 (1890).
17. Bayne, T., Hohwy, J. & Owen, A. M. Are There Levels of Consciousness? *Trends Cogn. Sci.* **20**, 405–413 (2016).
18. Fillingim, R. B. Individual differences in pain. *Pain* **158**, S11–S18 (2017).
19. Naci, L., Sinai, L. & Owen, A. M. Detecting and interpreting conscious experiences in behaviorally non-responsive patients. *Neuroimage* **145**, 304–313 (2017).
20. Poldrack, R. A. Can cognitive processes be inferred from neuroimaging data? *Trends Cogn. Sci.* **10**, 59–63 (2006).
21. Taylor, K. I., Moss, H. E., Stamatakis, E. A. & Tyler, L. K. *Binding Crossmodal Object Features in Perirhinal Cortex*. vol. 103 www.pnas.org/cgi/doi/10.1073/pnas.0509704103 (2006).
22. Taylor, K. I., Stamatakis, E. A. & Tyler, L. K. Crossmodal integration of object features: Voxel-based correlations in brain-damaged patients. *Brain* **132**, 671–683 (2009).
23. Feilong, M., Nastase, S. A., Guntupalli, J. S. & Haxby, J. V. Reliable individual

- differences in fine-grained cortical functional architecture. *Neuroimage* **183**, 375–386 (2018).
24. Anderson, Z., Gratton, C. & Nusslock, R. The Value of Hyperalignment to Unpack Neural Heterogeneity in the Precision Psychiatry Movement. *Biol. Psychiatry Cogn. Neurosci. Neuroimaging* **6**, 935–936 (2021).
 25. Anderson, Z., Turner, J. A., Ashar, Y. K., Calhoun, V. D. & Mittal, V. A. Application of hyperalignment to resting state data in individuals with psychosis reveals systematic changes in functional networks and identifies distinct clinical subgroups. *Aperture Neuro* **4**, 1–12 (2024).
 26. Charest, I., Kievit, R. A., Schmitz, T. W., Deca, D. & Kriegeskorte, N. Unique semantic space in the brain of each beholder predicts perceived similarity. *Proc. Natl. Acad. Sci.* **111**, 14565–14570 (2014).
 27. Yeshurun, Y. *et al.* Same Story, Different Story: The Neural Representation of Interpretive Frameworks. *Psychol. Sci.* **28**, 307–319 (2017).
 28. Marek, S. *et al.* Spatial and Temporal Organization of the Individual Human Cerebellum. *Neuron* **100**, 977–993 (2018).
 29. De Haas, B., Iakovidis, A. L., Schwarzkopf, D. S. & Gegenfurtner, K. R. Individual differences in visual salience vary along semantic dimensions. *Proc. Natl. Acad. Sci. U. S. A.* **116**, 11687–11692 (2019).
 30. Sylvester, C. M. *et al.* Individual-specific functional connectivity of the amygdala: A substrate for precision psychiatry. *Proc. Natl. Acad. Sci. U. S. A.* **117**, 3808–3818 (2020).